# Establishment of Repeated In Vitro Exposure System for Evaluating Pulmonary Toxicity of Representative Criteria Air Pollutants Using Advanced Bronchial Mucosa Models

**DOI:** 10.3390/toxics10060277

**Published:** 2022-05-24

**Authors:** Swapna Upadhyay, Ashesh Chakraborty, Tania A. Thimraj, Marialuisa Baldi, Anna Steneholm, Koustav Ganguly, Per Gerde, Lena Ernstgård, Lena Palmberg

**Affiliations:** 1Unit of Integrative Toxicology, Institute of Environmental Medicine, Karolinska Institutet, 171 77 Stockholm, Sweden; chakrabortyashesh958@gmail.com (A.C.); tat2151@cumc.columbia.edu (T.A.T.); 4baldi.marialuisa@gmail.com (M.B.); koustav.ganguly@ki.se (K.G.); per.gerde@ki.se (P.G.); lena.ernstgard@gmail.com (L.E.); 2Nynas AB, P.O. Box 10 700, 121 29 Stockholm, Sweden; anna.steneholm@nynas.com

**Keywords:** air pollution, repeated exposure, pulmonary toxicity, bronchial mucosa model

## Abstract

There is mounting evidence that shows the association between chronic exposure to air pollutants (particulate matter and gaseous) and onset of various respiratory impairments. However, the corresponding toxicological mechanisms of mixed exposure are poorly understood. Therefore, in this study, we aimed to establish a repeated exposure setting for evaluating the pulmonary toxicological effects of diesel exhaust particles (DEP), nitrogen dioxide (NO_2_), and sulfur dioxide (SO2) as representative criterial air pollutants. Single, combined (DEP with NO_2_ and SO_2_), and repeated exposures were performed using physiologically relevant human bronchial mucosa models developed at the air–liquid interface (bro-ALI). The bro-ALI models were generated using human primary bronchial epithelial cells (3–4 donors; 2 replicates per donor). The exposure regime included the following: 1. DEP (12.5 µg/cm^2^; 3 min/day, 3 days); 2. low gaseous (NO_2_: 0.1 ppm + SO_2_: 0.2 ppm); (30 min/day, 3 days); 3. high gaseous (NO_2_: 0.2 ppm + SO_2_: 0.4 ppm) (30 min/day, 3 days); and 4. single combined (DEP + low gaseous for 1 day). The markers for pro-inflammatory (IL8, IL6, NFKB, TNF), oxidative stress (HMOX1, GSTA1, SOD3,) and tissue injury/repair (MMP9, TIMP1) responses were assessed at transcriptional and/ or secreted protein levels following exposure. The corresponding sham-exposed samples under identical conditions served as the control. A non-parametric statistical analysis was performed and *p* < 0.05 was considered as significant. Repeated exposure to DEP and single combined (DEP + low gaseous) exposure showed significant alteration in the pro-inflammatory, oxidative stress and tissue injury responses compared to repeated exposures to gaseous air pollutants. The study demonstrates that it is feasible to predict the long-term effects of air pollutants using the above explained exposure system.

## 1. Introduction

Air pollution is an underestimated global issue with considerable consequences on the environment, economy, and human health [1,2]. In contrast to many other risks, exposure to outdoor air pollutions occurs during the entire human lifespan. Exposure is usually inevitable and involuntary, and its adverse health effects have been linked to increased mortality and also contribute to the risk of developing respiratory and cardiovascular diseases globally in low-, middle-, and high-income countries [1,3].

According to the European Environmental Agency [1], despite the action taken that aims to reduce pollutant emissions, the air quality is still poor and exceeds the established European Union limit values [2]. Moreover, regions such as East Asia (China Mainland) and South Asia (Bangladesh, India and Pakistan) have been classified as those with the highest levels of air pollution (2018 World Air Quality Report) with an unhealthy Air Quality Index [4,5,6,7].

The United States Environmental Protection Agency (US-EPA, 2021) defines the most common air pollutants as “Criteria Air Pollutants” (particulate matters, PM; sulfur dioxides, SO_2_; nitrogen dioxides, NO_2_; carbon monoxide, CO; ozone, O_3_ and Lead, Pb). These can be classified into the following two main groups: primary pollutants emitted directly to the atmosphere and secondary pollutants derived from precursor pollutants, transformed in the atmosphere through a chemical reaction. The effects of PM on health occur at levels of exposure currently being experienced by many people both in urban and rural areas and in developed and developing countries. Although, exposures in many fast-developing cities today are often far higher than in developed cities of comparable size. Epidemiological studies have demonstrated an association between exposure to urban ambient PM and an impairment of lung function and development of chronic inflammation, leading to asthma or COPD [8,9]. PM mainly consists of a complex mixture of solid and liquid particles of organic and inorganic substances suspended in the air. In urban areas, exhausts from diesel and gasoline vehicles constitute a major portion of PM and have been suspected of increasing the risk of cardio-pulmonary diseases. In a very recent study, Wongchung et al. (2021) demonstrated that diesel particulate matter (DPM) and oxides of nitrogen (NOx) are the emissions from diesel engines (compression ignition engines) of the most concern and are currently strictly regulated [10]. It has been already reported by several air pollution researchers that diesel exhaust (DE) emissions are the major source of PM in the urban environment that contribute a very high numbers of ambient particles and a large reactive surface area, as well as the potential to deposit throughout the lungs [11,12]. Moreover, depending on the size, PM can reach different parts of the lung, including the extra thoracic and upper respiratory tract (coarse particles, PM_10_) or deeper lung part, such as the alveoli (fine and ultrafine particles, PM_2.5_, PM_0.1_), triggering inflammation that can result in the development of chronic lung diseases, such asthma, chronic obstructive pulmonary disease (COPD) and/or chronic bronchitis CB, [13,14].

On the other hand, exposure to ambient gaseous air pollutants (O_3_, NO_2_, SO_2_, CO) has been linked to increased risks of mortality, respiratory and cardiovascular diseases. Evidence from epidemiological and in vivo studies suggests that both long-term and short-term exposure to gaseous air pollutants can induce oxidative damage in cells and lining fluids of the airways, resulting in subsequent inflammatory responses in the respiratory system [15,16]. Furthermore, chronic exposure to those pollutants has also been linked to enhanced sensitization and inflammatory responses within and beyond the lungs [17,18].

The effect of a single exposure to PM (e.g., DEP, carbon nanoparticles, etc.)-mediated pulmonary toxicity has already been well studied, but the combined effect of PM and gaseous air pollutants following repeated exposure are still not well established experimentally. This is primarily due to the lack of efficient test systems that mimic human inhalation exposure scenarios to air pollutants. Most of the pulmonary in vitro studies have been conducted using cell lines under submerged cell culture conditions, and thereby overlooking pulmonary physiology. Moreover, submerged cell culture systems lack the possibility of effective dose measurements. Particle properties, such as size, surface charge, solubility, transformation, or agglomeration state and chemical properties are altered in solution and are dependent on the composition of the cell culture medium [19,20,21]. Physiologically relevant in vivo-like in vitro multicellular lung models cultured at the air–liquid interface (ALI) are, therefore, becoming a realistic and efficient tool for lung toxicity testing and cell–cell interaction studies following repeated exposure to representative criteria air pollutants (aerosolized particles, DEP; gaseous air pollutants, SO_2_ + NO_2_) [22,23,24,25].

Hence, in this study, our primary aim was to establish and validate a repeated exposure setting and its toxic effects after exposure to air pollutants (both PM and gaseous substances) using physiologically relevant in vitro models. In the repeated exposure setting, bro-ALI models were exposed for 3 minutes to DEP (12.5 µg/cm^2^) for 3 consecutive days, while in separate experiments, models were exposed to gaseous air pollutants for 30 minutes using a combination of NO_2_ and SO_2_ in two different combinations of concentrations (low: 0.1 ppm NO_2_/0.2 ppm SO_2_ and high: 0.2 ppm NO_2_/0.4 ppm SO_2_) for 3 consecutive days. The selected concentrations of gaseous air pollutants were determined based on the air quality index reports from the most polluted cities in Europe, India and China in recent years [2,26,27,28]. A single exposure to the combination of DEP and gases (NO_2_ and SO_2_) was also performed. The markers of inflammatory, oxidative stress and tissue injury/repair responses were measured to assess the potential subacute toxicity of the air pollutants.

## 2. Materials and Methods

### 2.1. Bronchial Mucosa Model (Bro-ALI)

Bronchial mucosa models used in this study were developed with primary bronchial epithelial cells (PBEC) as described by Ji et al. [29]. The PBEC were harvested from healthy bronchial tissues obtained from donors in connection with lobectomy, following informed and written consent, and approval by the Swedish Ethical Review Authority. The detailed protocol and details of cellular differentiation (club cells, goblet cells, basal cells, ciliated cells, etc.) of the bro-ALI model have been described previously [23,24,25,26]. The PBECs used to develop the bro-ALI models were cultured at ALI conditions and are well characterized as described in our previous study [29]. In brief, PBECs were cultured under submerged conditions in a coated 75 mL petri flask. Cells were then cultured in complete PneumaCult^TM^ Ex Basal Medium with 50X Supplement (Catalog # 05008), Hydrocortisone Stock Solution 200x (Catalog # 07926) (STEMCELL Technologies, Cambridge, UK) and with 0.2% penicillin-streptomycin (Catalog # 15140122, Thermo-Fischer, Stockholm, Sweden). After two weeks, when cells had reached 90% confluence, the cells were trypsinized (Catalog#: 15090046, Thermo-Fischer, Stockholm, Sweden) and seeded into pre-coated 0.4 μm semiporous transwell inserts (catalog # 353180; BD Falcon™, San Diego, CA, USA) in a 12 well-plate. A total of 0.1 million of cells were seeded in each insert and cultured under submerged conditions with Pneumacult-EX complete medium as described above (1 mL was pipetted on the apical and 1 mL on the basal side of each insert). Once the cells were about 80% confluent, they were air-lifted by adding 1 mL of PneumaCult-ALI medium with 10X Maintenance Supplement (Catalog # 05001) (STEMCELL Technologies, Cambridge, UK), 0.5% hydrocortisone, 0.2% sodium heparin (2 mg/mL) (Catalog # 07980, STEMCELL Technologies, Cambridge, UK) and 0.2% penicillin-streptomycin only to the basal side of the inserts as described previously [22,29]. After two weeks under ALI conditions, the bro-ALI models were ready for exposure. The exposures were repeated, which included in total 3 to 4 donors (N = 3/4) and the number of technical replicates per experiment were 2 (n = 2).

### 2.2. Bro-ALI Model Exposure Systems

In the present study, the bro-ALI models were exposed by single and repeated exposure routes to different components of criteria air pollutants as described by the US-EPA, which are as follows:Particulate matter (PM): diesel exhaust particles (DEP).Gaseous air pollutants: nitrogen dioxides (NO_2_); sulfur dioxide (SO_2_).

#### 2.2.1. DEP Generation and Exposure System

DEP was generated and collected from a three-cylinder, 3.8 l tractor engine (Model 1113 TR; Bolinder-Munktell; an old model that was used earlier) at the Swedish Engine Test Center, Uppsala. The DEP soot was scraped from the Teflon-coated electrodes and stored in the dark at −20 °C. These DEP are well characterized and have been used earlier with bro-ALI models and combined with an aerosolized particle exposure system (Xpose*ALI*, 22). Aerosolized DEP was generated from small batches of dry particle powder using the high-pressure aerosol generator of the PreciseInhale exposure platform and triplicate model inserts were exposed to the aerosolized particles at the same time. Particles were loaded into the powder chamber of the PreciseInhale aerosol generator for each exposure cycle [22,23,29]. Compressed air of 100 bars were used to aerosolize the particles into the 300-mL holding chamber. The generated aerosol was pulled from the holding chamber with a constant flow rate (120 mL/min) and diverted into triplicate branch exposures at a flow rate of 10 mL/min per branch. To maintain viability of cells during the exposures, the holding chamber was humidified by covering the inside walls with wet filter papers. During the exposures, the inserts with the bro-ALI models were in contact with air-lifted medium from the bottom. Sham exposures (exposures with identical flow rate settings using only air and a clean exposure system) were carried out to control for potential viability effects on the model induced by the exposure system. Based on data from the study by Ji et al. in 2018 where we tested different doses, we decided to continue with an exposure for 3 minutes, corresponding to an exposure dose of 12.5 µg/cm^2^. The exposure of DEP was carried out for both single and repeated exposures.

#### 2.2.2. Gaseous Air Pollutants (Nitrogen Dioxides and Sulphur Dioxides) and Exposure System

The concentrations of the gases were determined based on the air quality index reports from the most polluted cities in Europe, India and China in recent years [2,28]. Based on the literature data on exposure levels, the bro-ALI models were exposed to two different combinations (low and high) of NO_2_ and SO_2_ levels; NO_2_ 0.1 ppm/SO_2_ 0.2 ppm (low doses) or NO_2_ 0.2 ppm/SO_2_ 0.4 ppm (high doses). The acute exposure of gases was performed by exposing cells to the above-mentioned concentrations for 30 minutes, while the chronic exposure was carried out by exposing cells for 30 mins three days in a row, once per day. The above-mentioned concentrations for the combination of gases were generated in a 30 L tedlar bag diluted with room temperature air. The exposure of the bronchial mucosa models to the combination of gases was carried out in glass exposure chambers connected to the Tedlar bag with a flow rate of 0.5 L/min, as described previously [24,25]. The exposure chamber was placed in a hot water bath to maintain the temperature at 37 °C inside the chamber. A total of 200 µL of sterile water was added to the bottom side of the exposure chamber for achieving humidity between 50–60%. In this experimental setup, room temperature clean air was used as a control exposure setting (Sham).

#### 2.2.3. Repeated Exposures to Diesel Exposure Particles and Gaseous Air Pollutants

To mimic the ambient exposure scenario, we developed methods for repeated exposures to particles (DEP) and gaseous air pollutants (NO_2_ in combination with SO_2_) for 3 days. Cell culture medium was collected and replaced with fresh ALI medium 24 h after each exposure, before the following exposure was carried out. Hence, the protein concentrations measured in this study represent the total secretion of protein in the basal medium 24 h after each exposure.

In the repeated exposures, the bro-ALI models were exposed for 3 min per day to aerosolized DEP (12.5 µg/cm^2^) using the PreciseInhale aerosol generator as mentioned above for 3 consecutive days. Both basal medium and cell samples were collected at three different time points as shown in Figure 1, following repeated exposure to DEP, which included the following: exposure 1 (24 h), N = 3 donors and n = 2 per donors and, exposure 2 (48 h), N = 3 donors and n = 2 per donor’s samples collected, and Exposure 3 (72 h), N = 3 donors and n = 2 per donors. Basal media were collected for protein release and cells were collected for transcript analysis.

In the same way as DEP, in a separate exposure system, thr bro-ALI models were exposed to gaseous air pollutants for 3 consecutive days and 30 min/day using the combination of NO_2_ and SO_2_ in two different concentrations (low: 0.1 ppm NO_2_/0.2 ppm SO_2_ and high: 0.2 ppm NO_2_/0.4 ppm SO_2_). Both basal medium (BM) and cell samples were collected at three different time points as shown in Figure 1, following repeated exposure to gaseous pollutants, which included the following: exposure 1 (24 h), N = 3 donors and n = 2 per donors collected; exposure 2 (48 h), N = 3 donors and n = 2 per donors; exposure 3 (72 h), N = 3 donors and n = 2 per donors. Both basal medium and cells were collected for further analysis. BM were collected for protein release and cells were collected for transcript analysis.

#### 2.2.4. Single Combined Exposure to DEP and Gaseous (NO_2_ and SO_2_) Air Pollutants

To mimic the in vivo exposure situation, we wanted to establish methods to mimic a combined exposure scenario. In the combined exposure setting, the bro-ALI models were exposed to gaseous air pollutants (low exposure dose used above: 0.1 ppm NO_2_ in combination with 0.2 ppm SO_2_) for 30 min, immediately followed by 3 min exposure to DEP (12.7 µg/cm^2^, Figure 1). After both exposures were carried out, the bro-ALI models were incubated for 24 h before taking both BM and cells for further analysis of the protein and RNA levels of inflammatory, oxidative stress and tissue injury/repair markers (see below).

### 2.3. Sample Analysis

After single and repeated exposures, both cell culture media and cells were collected and stored at −80 °C for further analysis. The BM was collected after 24 of every exposure (e.g., exposure 1; exposure 2, exposure 3; Figure 1) for analysis of cell viability, tissue injury/repair and inflammatory markers at protein levels. Similarly, cells were collected for mRNA expression analysis for inflammation, oxidative stress, and tissue injury/repair responses after overnight incubation of every exposure. By this design, the mRNA expression after the repeated exposures reflects the cumulative effect, whereas the protein levels in the cell culture media represent production following every 24 h of exposure.

#### 2.3.1. Cell Viability: Lactate Dehydrogenase (LDH) Assay

The cell viability of the bro-ALI models following exposure to clean air (sham) as well to air pollutants (DEP, gases and combined exposure) was assessed by measuring LDH release in BM collected 24 h after incubation of every exposure (e.g., exposure 1; exposure 2, exposure 3; Figure 1). The LDH measurement was analyzed using the Pierce™ LDH Cytotoxicity Assay Kit and performed in accordance with the manufacturer’s instructions (Thermo-scientific, MA, USA catalog no: 88953). In short, 50 µL of BM collected after the exposures were transferred to the 96 well-plates, followed by the addition of 50 µL reaction mixture to each sample well and then incubated for 30 min in dark, at room temperature. After the incubation, the LDH release was measured using a plate reader at 490 nm and 680 nm wavelength [24,30].

#### 2.3.2. Secreted Protein Concentration Measurements by ELISA

After single or repeated exposures of bro-ALI to both DEP and gases, the BM was collected 24 h after the incubation of every exposure (Figure 1) to air pollutants and was used to evaluate chemokine (IL8) and tissue injury (MMP 9) marker release. The IL8 protein levels were determined using the Duoset IL8 ELISA KIT (R & D Systems, Minneapolis, MN, US, Catalog # DY208) and the MMP9 protein levels were measured using the MMP9 Duoset ELISA KIT (R & D Systems, Minneapolis, MN, US, Catalog # DY911) and carried out according to the manufacturer’s instructions (R&D Systems, Sweden). The detection limit for both IL8 and MMP 9 was 15.65 pg/mL. The measurement for both MMP9 and IL8 was determined using a plate reader at 450 nm and 540 nm.

#### 2.3.3. Transcript Expression Analysis by Real-Time Quantitative Reverse Transcription-PCR (RT-qPCR)

Total RNA extractions and RT-qPCR analysis following the repeated exposures of the bro-ALI models to DEP and gases were performed using the RNeasy Mini Kit (Qiagen# 74004, Hilden, Germany). The concentration and the purity of the extracted RNA were determined using the Nanodrop 1000 (Thermo Scientific, MA, USA) 260/280 ratio. A total of 1 μg of mRNA was reverse transcribed to generate complementary DNA (cDNA) using the High-Capacity RNA to cDNA Kit (Life technologies, Paisley, UK) using a thermal cycler (MycyclerTM, Biorad, Hercules, CA, USA). RT-qPCR was performed using the AB 7500 System (Thermo Scientific, MA, USA). The 20 μL qRT-PCR reaction mix consisted of 10 μL Fast SYBR^®^ Green Master Mix (Life technologies, Paisley, UK), 200 nmol of each primer, 5 ng cDNA, and nuclease free water. Beta actin (*ACTB*) was used as the reference control. The expression of pro-inflammatory (interleukin: (*IL8*, *IL6*) and tumor necrosis factor alpha (*TNFα*)), oxidative stress (Glutathione S-Transferase Alpha 1 (GSTA1), heme oxygenase 1 (HMOX1) and superoxide dismutase 3 (SOD3)) and tissue injury/repair genes (matrix metallopeptidase 9 (*MMP9*) and tissue inhibitor matrix metalloproteinase 1 (*TIMP1*)) were quantified as a fold change following normalization with *ACTB* and sham of bro-ALI. The results were calculated as 2-ΔCt (ΔCt = Ct (gene of interest) - Ct (beta actin). The primer pair sequences of the investigated genes analyzed the genes as described previously [22,23,24,25].

#### 2.3.4. Statistics

The results are expressed as median and interquartile ranges (25th–75th percentiles) and a non-parametric statistical analysis was performed as previously described [31]. Within each group (sham versus exposure), the comparisons between different exposed groups were assessed by the Friedman test and followed by the Wilcoxon signed rank t test as a post hoc test. In all the tests, p values below 0.05 were considered significant. All the data were analyzed using the STATISTICA9 software (StatSoft, Inc. Uppsala, Sweden).

## 3. Results

### 3.1. Effects of DEP and Gaseous Air Pollutants Exposure on Cell Viability (Lactate Dehydrogenase Assay)

The LDH assay did not exhibit any significant alteration in cell toxicity after the exposure of the bro-ALI models to clean air (sham), repeated doses of DEP, or gases or after single combined exposure to both DEP and gases (Figure 2a–c). None of the doses used in this study were cytotoxic. 

### 3.2. Repeated DEP Exposure Induced Pro-Inflammatory, Oxidative Stress and Tissue Injury/Repair Effect on Bro-ALI

#### 3.2.1. Pro-Inflammatory Effects

In bro-ALI, IL8 secretion was significantly decreased at 48 and 72 h post-exposure to DEP (12.5 μg/cm^2^), compared to the sham and following exposure 1 (24 h) to DEP (Figure 3).

The transcript expression of the pro-inflammatory (*IL8, IL6* and *TNF*, Figure 4a–c) and oxidative stress markers (GSTA1, HMOX1 and SOD3, Figure 4d–f) were assayed in the bro-ALI models at 24 h, 48 h and 72 h post-exposure to sham and aerosolized DEP. Compared with sham, the transcript expression of *IL8* showed time-dependent increased expression at 24 h, 48 h and 72 h post-exposure to DEP; however, statistically significant increased expression was detected only 48 h after DEP exposure (Figure 4a). The expression of *IL6* was significantly enhanced 72 h after repeated DEP exposure compared with sham and 24 h following first exposure to DEP (Figure 4b). Interestingly, the transcript expression of *TNF* significantly increased 24 h and 48 h following exposure to DEP compared to sham, and repeated exposures to DEP showed the continuous induction of TNF expression until 72 h post-exposure, although not significant (Figure 4c), which is the same as *IL8*. Similarly, *GSTA1* significantly increased both at 24 h and 72 h after DEP exposure, compared to sham exposure (Figure 4d). Furthermore, the expression of *HMOX1* showed time-dependent increased expression both at 24 and 48 h post-exposure to DEP compared to the sham-exposed bro-ALI models (Figure 4e). Similarly, the time-dependent increased expression of *SOD3* was observed both at 48 and 72 h post DEP-exposure compared to sham, although only significant after 48 h (Figure 4f).

#### 3.2.2. Tissue Injury/Repair

In bro-ALI, the tsecretion of MMP9 in BM was significantly reduced 24 h after exposure 3 (72 h after first exposure) compared to sham (Figure 5), while the secretion of MMP9 remained unchanged 24 h after exposure 1 (24 h) and 2 (48 h). Significant induction of *MMP9* expression was observed both 24 and 72 h post DEP exposure (Figure 6a,b) compared to sham (Figure 6a). The transcript expression of *TIMP1* in bro-ALI remained unaltered at all time points following repeated exposure to DEP (Figure 6b).

### 3.3. Repeated Gas (NO_2_ and SO_2_) Exposure Induced Pro-Inflammatory, Oxidative Stress and Tissue Injury/Repair Effect on Bro-ALI

The transcript expression of pro-inflammatory (Appendix A), oxidative stress (Appendix A), and tissue injury/repair markers (Appendix A) were measured following repeated exposure to low and high concentration of gaseous (NO_2_ and SO_2_) air pollutants. The expression of tissue injury marker MMP9 significantly reduced both at 48 and 72 h (Appendix A) after exposure to high gas compared to sham. Although similar responses were observed with most of the other markers (pro-inflammatory, oxidative stress or tissue repair markers), none of these markers were statistically significant.

Similarly, the concentrations of IL8 and MMP9 protein in BM significantly reduced mostly at all time points (24 h after exposure to 3 consecutive time points: 24, 48 and 72 h, Appendix A) after both low and high gas exposure, except for the significantly increased concentration of IL8 (Appendix A), which was measured at 24 after high gas exposure.

### 3.4. Single Combined Exposures (DEP with NO_2_ and SO_2_) Induced Pro-Inflammatory, Oxidative Stress and Tissue Injury/Repair Effect in Bro-AL

Figure 7 and Figure 8 show the transcript expression of pro-inflammatory, oxidative stress and tissue injury/repair markers following single combined exposure to DEP and gaseous (NO_2_ and SO_2_) air pollutants. The pro-inflammatory marker (*TNF*: Figure 7c) and oxidative stress markers (*GSTA1* and *SOD3*: Figure 7d,f) showed significantly increased expression 24 h post exposure. The transcript expression of all the other markers (inflammatory: *IL6*, oxidative stress: *HMOX1*) also showed induced expression following single combined exposure to DEP and gaseous air pollutants, although the alterations were not statistically significant. However, the significantly induced expression of both tissue injury (*MMP9*) and repair (*TIMP1*) markers 24 h after single combined exposure to DEP and gases (Figure 8a,b) was observed. This finding is comparable with the increased concentration of MMP9 protein levels after exposure, compared to sham-exposed bro-ALI (Figure 9b). However, the concentration of IL8 at the protein level in BM remains unchanged 24 h after single combined exposure to DEP and gases (Figure 9a).

## 4. Discussion and Conclusions

Exposure to air pollutants is a potential risk factor for the development or exacerbation of many respiratory diseases, such as COPD, particularly in the most susceptible individuals. The health of susceptible and sensitive individuals can be impacted even on low air pollution days. There are many pollutants that are major contributors to the onset of health problems in humans [31,32,33]. Among them, PM-mediated adverse health effects have already been well studied in epidemiology and have also been used in vivo and/or in vitro models to explain the risk of size and chemical composition of PM [23,28]. For example, fine (PM_2.5_) and ultrafine (PM_0.1_) PM, due to their small size and large reactive surface area, can penetrate deeply into the lungs, reaching the alveoli and potentially translocating in the bloodstream, which may cause both local (lung) and systemic (blood, cardiovascular system) inflammation, as well the onset of cardio-pulmonary impairments [34,35,36]. Furthermore, these air pollutants (PM) have been accompanied by an increase in the levels of other pollutants, such as nitrogen oxide, sulfur dioxide, volatile organic compounds (VOCs), dioxins, and polycyclic aromatic hydrocarbons (PAHs), which are all considered as air pollutants that are harmful to humans [17,36,37]. However, only few epidemiological, in vivo and/or in vitro studies are available that show the combined effect of PM and gaseous components of air pollutants on the exacerbation of respiratory diseases. Both short- and long-term exposures to air pollutants are closely related to either onset or exacerbation, including cough, shortness of breath, wheezing of patients with asthma or COPD and high rates of hospitalization due to chronic local (lung) or systemic inflammation (blood). The long-term effect of air pollutants has been studied mostly using in vivo studies [38,39,40] but the underlying mechanisms are far from being understood. As epithelial cells are one of the first targets of PM and as the epithelium is recognized as a dynamic barrier that determines the fate of the bronchial wall [40], suitable in vitro models are needed to investigate the long-term responses of the bronchial epithelium to air pollutants. 

Therefore, the primary aim of this experimental setup was to investigate the principal activation of the three main cellular mechanisms (inflammation, oxidative stress and tissue injure/repair) behind the development of the most common chronic airway diseases (e.g., chronic bronchitis, COPD) induced by air pollutant (NO_2_, SO_2_ and DEPs) exposures. In this study, we have analyzed the air pollutant-mediated toxicity response following repeated exposure to diesel exhaust particles (DEP) or gases (NO_2_ and SO_2_) and in a single combined exposure to particles followed by exposures to gaseous air pollutants, using physiologically relevant bronchial mucosa models, including human primary bronchial epithelial cells grown at ALI.

The cell viability assessment following repeated DEP exposure showed that the selected DEP dose (12.5 µg/cm^2^) did not induce any significant cytotoxicity even 72 h after exposure. Instead, more than 90% cells remained viable, which is comparable with a previous study by Jie et al. [22], following single exposure to DEP. A similar cell viability response was detected following both repeated exposures to gaseous pollutants, as well as after single exposure to DEP combined with exposure to gaseous pollutants. However, the main limitation of this study was the higher risk of cell infection following the repeated and combined exposures to both DEP and gaseous components of air pollutants that might be due to contamination during the repeated exposures outside the incubator.

The pro-inflammatory (*IL8, IL6* and *TNF*) response induced by repeated exposure to DEP in the present study is comparable with the transcript expression of pro-inflammatory markers detected in our previous study following a single exposure to DEP, as well as following the repeated exposure to PM reported by several researchers [41,42,43]. Auger et al. [42] reported that the exposure of human nasal epithelial cells cultured at ALI to DEP and PM_2.5_ (10–80 μg/cm^2^) for 24 h stimulated both IL-8 and amphiregulin (ligand of EGFR) secretion exclusively towards the basal compartment. In contrast, there was no interleukin-1b(IL-1b) secretion and only weak non-reproducible secretion of TNF-α. IL-6 were detected only when the cells were exposed to PM_2.5_. In the present study, the release of chemokines (IL8) was significantly reduced following repeated (48 and 72 h) exposure to DEP, compared to sham. Veronesi et al. [44] reported reduced levels of IL6 release from BEAS-2B cells following exposure to residual oil fly ash particles. The induced expression of several oxidative stress response-related markers (*HMOX1*, *GSTA1* and *SOD3*) indicates direct evidence that repeated DEP exposure mediates reactive oxygen species generation. Several in vitro studies have identified macrophages and bronchial epithelial cells as important cellular targets for air pollutants (e.g., PM) in the lungs [22,45,46]. Exposure of these cells to ambient PM and other air pollutants (gaseous, chemical compounds) can induce the generation of ROS and oxidative stress, which can result in increased cytokine and chemokine production [15,47]. These include increased TNF-α, IL-8 and IL-6 production in macrophages and in bronchial epithelial cells in association with PM-induced oxidative stress [48,49]. Similar responses were reported by Auger et al. [42] and in this study, the authors demonstrated that airway epithelial cells exposed to particles augment the local inflammatory response in the lung but cannot alone initiate a systemic inflammatory response. The similarly increased expression of MMP9 plays an important role for a variety of homeostatic functions and elicits repair responses as balance mechanisms in many chronic lung diseases, such as COPD [50], while unaltered TIMP, which is an inhibitor of MMP9, further supports the alteration of extracellular matrix homeostasis following repeated exposure to DEP.

Repeated exposure to gaseous air pollutants did not exhibit profound pro-inflammatory, oxidative stress or tissue injury responses as compared with other studies, where both NO_2_ and SO_2_ concentration used for exposure were much higher. In 2006, Bakand et al. [51] reported that cell viability was significantly reduced after the exposure of human A549 lung-derived cells to NO_2_ (2.5–15 PPM) or SO_2_ (10–200 ppm). Similarly, Knorst et al. [52] showed that the ability of alveolar macrophages to release TNF-a and IL-1b was significantly impaired following a 30 min exposure to 5 ppm SO_2_ exposure. 

Moreover, our study showed that the responses (pro-inflammatory, oxidative stress and tissue injury/repair) following the single combined exposure to DEP and gaseous pollutants were the same as those detected following the repeated exposure to only DEP. The significantly increased secretion of MMP9 only after the combined exposure to DEP and gaseous air pollutants, compared to the repeated exposure either to DEP or gaseous air pollutants, may indicate an additive cellular response due to the combined exposure of particles and gaseous air pollutants [53]. Furthermore, the combined exposure also significantly increased the expression of TIMP1, although no significant change was observed 24 h after the single exposure. It has already been well established that DEP exposure induces a pro-inflammatory response in bronchial epithelial cells due to their organic content, which also is responsible for the generation of reactive oxygen species [54]. Concerns regarding the combined toxicity of PM and other air pollutants for human health has been raised by many researchers [55,56]. The cellular response triggered by air pollutants (PM or SO_2_, NO_2_ etc.) is usually assessed by focusing on a particular air pollutant, while their interaction with co-contaminants can have a deep impact, either positively or negatively. However, the interactions between the contaminants and their resulting combined cellular toxicity or molecular response on in vitro or in vivo studies are often overlooked [57].

To conclude, in this study, we developed an experimental strategy that made it possible for the repeated exposure to air pollutants (particles and gases) of in vitro models with human primary bronchial epithelial cells grown at ALI to measure air pollutant-mediated long-term lung toxicity. Repeated particle exposures produced concomitant pro-inflammatory, oxidative stress and tissue injury responses. The mechanisms involved in these long-term effects following exposure to gaseous air pollutants need to be further evaluated. The study showed that the exposure system and the advanced in vitro models are compatible to perform repeated air pollution-mediated toxicity studies. Taken together, the findings of this study clearly indicate the importance of evaluating the effect of combined exposure to different air pollutants and the associated biological responses. Additionally, the establishment of these experimental systems to reduce animal usage warrants further investigation. However, further studies with reliable data will result in better and more reliable air pollution-mediated health risk assessments.

## Figures and Tables

**Figure 1 toxics-10-00277-f001:**
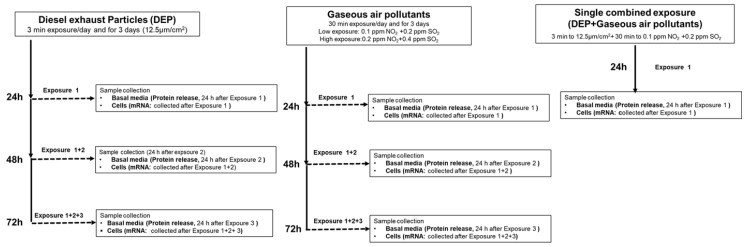
Schematic presentation of the overall experimental design. Basal medium was collected and replaced with fresh air–liquid interface medium 24 hour (h) after each exposure, before the following exposure was carried out.

**Figure 2 toxics-10-00277-f002:**
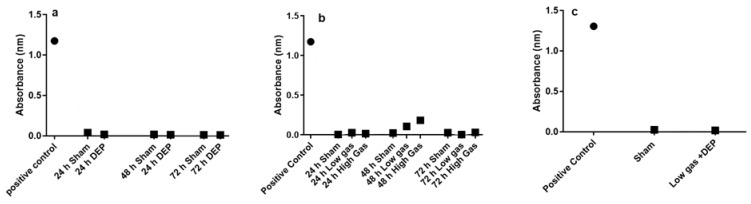
Cytotoxicity following repeated exposure to diesel exhaust particles (DEP: 12.7 µg/cm^2^; (**a**), low (NO_2_: 0.1 ppm; SO_2_: 0.2 ppm) and high (NO_2_: 0.2 ppm; SO_2_: 0.4 ppm; (**b**) and single combined exposure to DEP and low gas (**c**). The colorimetric lactate dehydrogenase assay used to measure the cytotoxic effect on bronchial mucosa models developed at the air-liquid interface (bro-ALI: N = 3, n = 2/donor) following overnight incubation of every exposure, positive control (kit provided).

**Figure 3 toxics-10-00277-f003:**
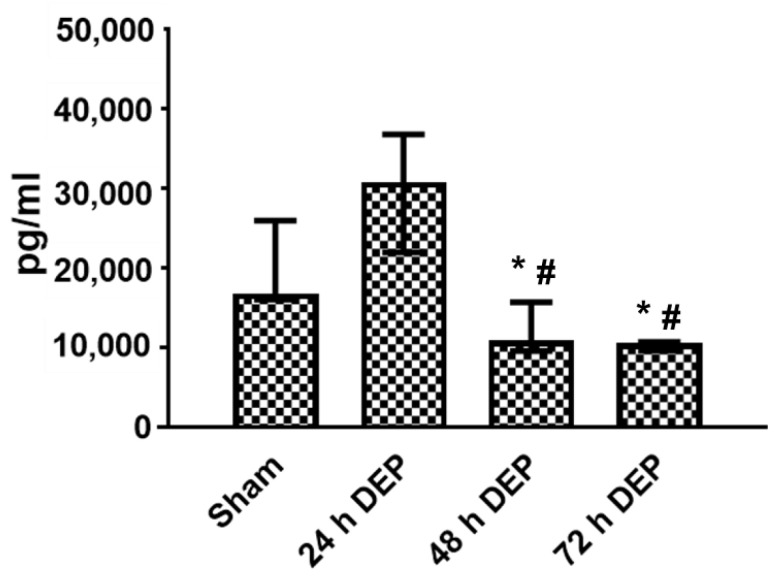
Release of IL8 in basal medium following repeated exposure to diesel exhaust particles (DEP: 12.7 µg/cm^2^). Concentration of IL8 in bronchial mucosa models developed at the air-liquid interface (bro-ALI) following incubation for 24 hours (h) after 3 min exposure to sham (control: clean air exposure) and DEP for 3 time points (24, 48 and 72 h). Data presented as median and 25th–75th percentiles (N = 3 donors and n = 2 replicates/donor). *p* < 0.05 *: significant compared to sham, #: significant compared to exposed at 24 h DEP.

**Figure 4 toxics-10-00277-f004:**
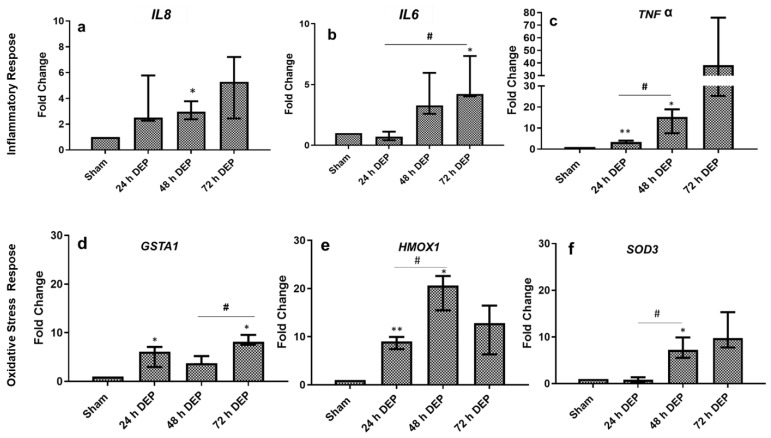
mRNA expression of inflammatory and oxidative stress markers following repeated exposure to diesel exhaust particles (DEP:12.7 µg/cm^2^). Fold change in inflammatory *(IL8, IL6, and TNFα (***a**–**c**)) and oxidative stress (*GSTA1, HMOX1, and SOD3* (**d**–**f**)) responses in bronchial mucosa models developed at the air–liquid interface (bro-ALI) following incubation for 24 hours (h) after 3 min exposure to sham (control: clean air exposure) and DEP for 3 time points (24, 48 and 72 h). Data presented as median and 25th–75th percentiles, fold change = 2-ΔCt of models / 2-ΔCt of sham-exposed bro-ALI (N = 3 donors and n = 2 replicates/donor). *p* < 0.05/0.01 */**: significant compared to sham; #: significant compared to exposed.

**Figure 5 toxics-10-00277-f005:**
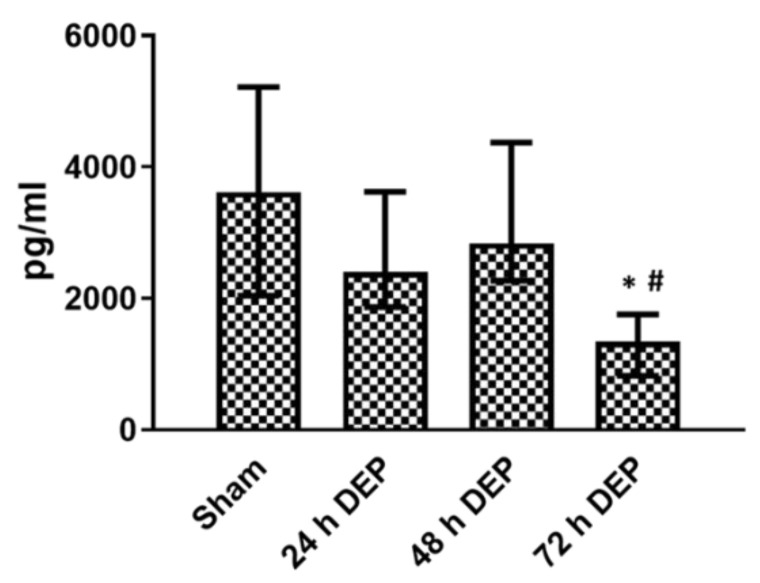
Release of MMP9 in basal medium following repeated exposure to diesel exhaust particles (DEP: 12.7 µg/cm^2^). Concentration of MMP9 in bronchial mucosa models developed at air–liquid interface (bro-ALI) following incubation for 24 hours (h) after 3 min exposure to sham (control: clean air exposure) and DEP for 3 time points (24, 48 and 72 h). Data presented as median and 25th–75th percentiles (N = 3 donors and n = 2 replicates/donor). *p* < 0.05 *: significant compared to sham, #: significant compared to exposed at 24 h.

**Figure 6 toxics-10-00277-f006:**
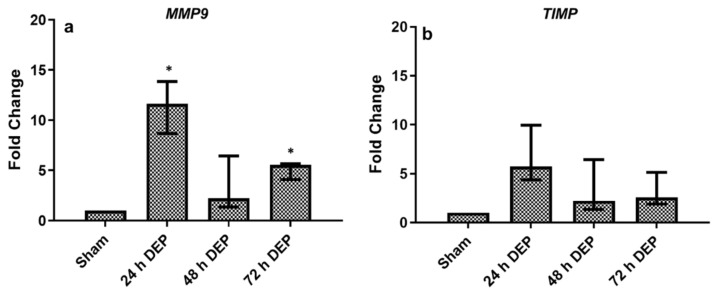
mRNA expression of tissue injury/repair markers following repeated exposure to diesel exhaust particles (DEP:12.7 µg/cm^2^). Fold change in *MMP9*, and *TIMP* in bronchial mucosa models developed at air–liquid interface (bro-ALI) following incubation for 24 hours (h), after 3 min exposure to sham (control: clean air exposure) and DEP for 3 time points (24, 48 and 72 h). Data presented as median and 25th–75th percentiles, fold change =2-ΔCt of models/2-ΔCt of sham-exposed bro-ALI. N = 3 donors and n = 2 replicates/donor). *p* < 0.05 *: significant compared to sham.

**Figure 7 toxics-10-00277-f007:**
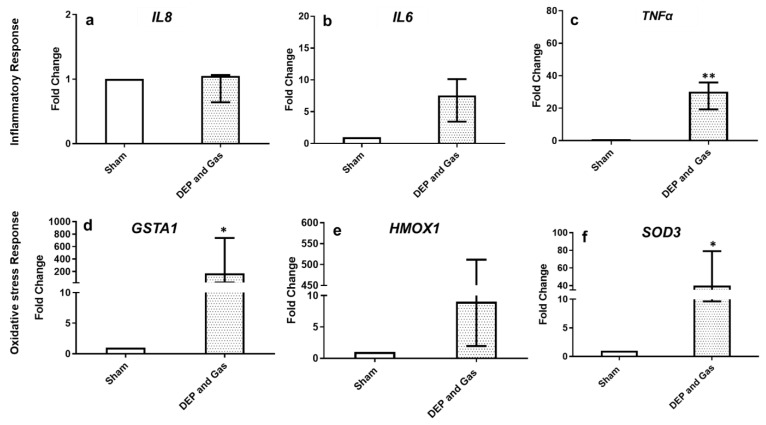
mRNA expression of inflammatory and oxidative stress markers following single combined exposure to diesel exhaust particles (DEP:12.7 µg/cm^2^) and gaseous air pollutants (0.1 ppm NO_2_, 0.2 ppm SO_2_). Fold change in inflammatory (*IL8, IL6, and TNFα* (**a**–**c**)) and oxidative stress [GSTA1, HMOX1, and SOD3 (**d**–**f**)] in bronchial mucosa models developed at air–liquid interface (bro-ALI) following incubation for 24 h after exposure to sham (control: clean air exposure) and combined air pollutants. Data presented as median and 25th–75th percentiles, fold change = 2-ΔCt of models/2-ΔCt of sham-exposed bro-ALI. N = 3 donors and n = 2 replicates/Donor. *p* < 0.01 **: significant compared to sham. *p* < 0.05 *: significant compared to sham.

**Figure 8 toxics-10-00277-f008:**
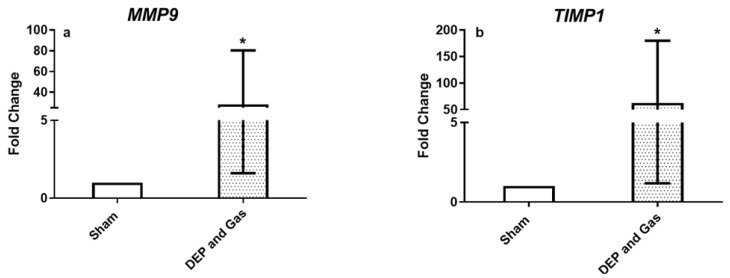
mRNA expression of tissue injury/repair markers following single combined exposure to diesel exhaust particles (DEP:12.7 µg/cm^2^) and gaseous air pollutants (0.1 ppm NO_2_, 0.2 ppm SO_2_). Fold change in *MMP9* (**a**), and *TIMP1* (**b**) in bronchial mucosa models developed at air–liquid interface (bro-ALI) following incubation for 24 h after exposure to sham (control: clean air exposure) and combined air pollutants. Data presented as median and 25th–75th percentiles, fold change = 2-ΔCt of models/2-ΔCt of sham-exposed bro-ALI. N = 3 donors and n = 2 replicates/donor). *p* < 0.05 *: significant compared to sham.

**Figure 9 toxics-10-00277-f009:**
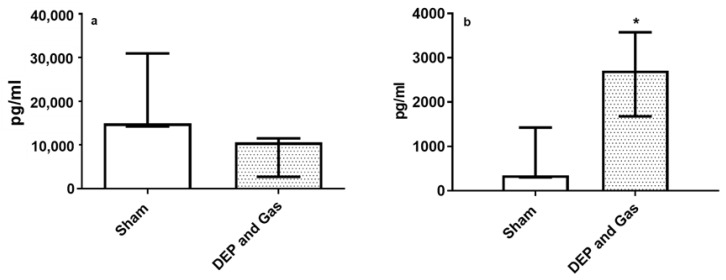
Release of IL8 and MMP9 in basal medium following repeated exposure to diesel exhaust particles (DEP:12.7 µg/cm^2^) and gaseous air pollutants (0.1 ppm NO_2_, 0.2 ppm SO_2_). Concentration of IL8 (**a**) and MMP9 (**b**) in bronchial mucosa models developed at air–liquid interface (bro-ALI) following incubation for 24 h after exposure to sham (control: clean air exposure) and combined air pollutants. Data presented as median and 25th–75th percentiles (N = 3 donors and n = 2 replicates/donor). *p* < 0.05 *: significant compared to sham.

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
