# Peer review of "Establishment of Repeated In Vitro Exposure System for Evaluating Pulmonary Toxicity of Representative Criteria Air Pollutants Using Advanced Bronchial Mucosa Models"

_toxics, 2022, doi:10.3390/toxics10060277_

Round 1

Reviewer 1 Report

Overall, the manuscript is well-written, and results are clear to understand. However, there are some minor points to clarify.

  1. Lines 91-93, it would be useful to provide what this paper adds compared to previous ones that seem to address the same topic using bronchial mucosa models.
  2. Please provide references for the combinations of concentrations (lines 100-102).
  3. What do you mean by sub-chronic exposure?
  4. The study design is not very clear. For example, I do not understand when PBEC was collected. Was this harvested from healthy bronchial tissue?

Author Response

To

The Editorial Board Members, Toxics,

Dear Madam/Sir,

Thanks for the comprehensive review of our manuscript titled “Establishment of sub-chronic repeated in vitro exposure system for evaluating pulmonary toxicity of representative criteria air pollutants using advanced bronchial mucosa models. (Manuscript ID: toxics-169139)”.

We have now addressed all the comments and suggestion by the reviewers. Below is the point-to-point reply to the reviewer’s comments. The edited text has been highlighted in red within the manuscript body. The newly added references and the reference numbers are in red and highlighted with yellow.

We do believe you will find the manuscript significantly improved and suitable for publication.

Sincerely

Swapna Upadhyay

Reviewer 1: Overall, the manuscript is well-written, and results are clear to understand. However, there are some minor points to clarify.

Thanks for appreciating our research.

  1. Lines 91-93, it would be useful to provide what this paper adds compared to previous ones that seem to address the same topic using bronchial mucosa models.

We have added as suggested (lines #92-96).2. Please provide references for the combinations of concentrations (lines 100-102)

References regarding the concentration of gaseous pollutants on lines 104-106 are now included (References # 26-28).

  1. What do you mean by sub-chronic exposure?

In the present study we have performed exposure of cell models to DEP for 3min/day and for 3 consecutive days. Similarly, exposure of cell models to gaseous air pollution were carried for 30 min/day and for 3 consecutive days. Therefore, we have used the term repeated or sub chronic exposure, though sub-chronic exposure of In Vitro models to any toxicants/xenobiotics is normally indicate much longer period or duration. Hence, we have revised the text and only “repeated exposure” has now been mentioned in the text to make it clearer.

  1. The study design is not very clear. For example, I do not understand when PBEC was collected. Was this harvested from healthy bronchial tissue?

Based on the reviewers’ suggestion we have now added the following details in the text (lines 110-114). The bro-ALI model was developed using PBEC from 3-4 donors (N) with 3 technical replicates (n) from each donor. The PBEC were harvested from healthy bronchial tissues obtained from donors in connection with lobectomy following informed and written consent, and approval by the Swedish Ethical Review Authority. The detailed protocol and details of cellular differentiation (club cells, goblet cells, basal cells, ciliated cells, etc.) of the bro-ALI model have been described previously23-26. The cells used in this study are well characterized and have been used in connection with several other projects

Reviewer 2 Report

Upadhyay et al. aimed to estimate the sub-chronic in vitro exposure system for 2 evaluating pulmonary toxicity using advanced bronchial mucosa models. They found that repeated exposure to DEP or combined (DEP + low gaseous) exposure induced significant alteration of pro-inflammatory, oxidative stress and tissue injury responses. The idea and the results of the study are interesting and impressed. I think this study warrants to be published. Here I addressed some concerns:

Major concern

  1. There are too many figures. I think that authors should reduce the number of graphs, and combine some figures, such as figure 4 and 5, figure 6 and 7.

  1. The objective of the study was to evaluate the combination effects of DEP with/without gaseous pollutants. However, I just saw the results of “DEP compared with control” and “DEP + gaseous pollutant compared with control”. Why didn’t the authors compare the results of “DEP” and “DEP + gaseous pollutant”?

  1. In figure7, MMP9 expression of DEP exposure groups was statistically significant different with control groups in “24 hours” and “72 hours”, but not 48 hours. Could you explain the finding? Similarly, in figure 5, the up regulation of HMOX1 and SOD3 was observed in “48 hours” groups, but not “72 hours” groups. Please explain the findings.

  1. The inflammatory markers chosen in the DEP group were IL-8, IL-6 and TNF-α, but the markers used in the “DEP + gaseous pollutant” groups were CXCL8 and CXCL6. Why didn't they use the same markers and compare them together?

Minor concern

  1. The figures should labeled as “A”, “B” and so on if there were more than one figure in “figure 4”, “figure 5” and so on.

  1. CXCL8, CXCL6 were up-regulated in “DEP + gas” group. However, I did not read the relative discussion. Please add some discussion.

3. Please explain why you select DEP as “particle” exposure, not PM collected from road or industrial area? 

Author Response

To

The Editorial Board Members, Toxics,

Dear Madam/Sir,

Thanks for the comprehensive review of our manuscript titled “Establishment of sub-chronic repeated in vitro exposure system for evaluating pulmonary toxicity of representative criteria air pollutants using advanced bronchial mucosa models. (Manuscript ID: toxics-169139)”.

We have now addressed all the comments and suggestion by the reviewers. Below is the point-to-point reply to the reviewer’s comments. The edited text has been highlighted in red within the manuscript body. The newly added references and the reference numbers are in red and highlighted with yellow.

We do believe you will find the manuscript significantly improved and suitable for publication.

Sincerely

Swapna Upadhyay

Reviewer 2: Upadhyay et al. aimed to estimate the sub-chronic in vitro exposure system for 2 evaluating pulmonary toxicity using advanced bronchial mucosa models. They found that repeated exposure to DEP or combined (DEP + low gaseous) exposure induced significant alteration of pro-inflammatory, oxidative stress and tissue injury responses. The idea and the results of the study are interesting and impressed. I think this study warrants to be published. Here I addressed some concerns:

Thanks for appreciating our research.

Major concern

  1. There are too many figures. I think that authors should reduce the number of graphs, and combine some figures, such as figure 4 and 5, figure 6 and 7.

Based on the reviewers’ suggestion we have combined Figures 4 and 5 as Figure 5, and Figures 8-9 as Figure 7.

  1. The objective of the study was to evaluate the combination effects of DEP with/without gaseous pollutants. However, I just saw the results of “DEP compared with control” and “DEP + gaseous pollutant compared with control”. Why didn’t the authors compare the results of “DEP” and “DEP +gaseous pollutant”?

In this manuscript we have presented the results of each exposure study separately because exposure system used for particles (DEP) and gaseous elements (SO2+NO2) are having different set-up as described in the methods section. Since all these exposure systems are quite different, and most of the markers (pro-inflammatory, oxidative stress or tissue repair markers) following exposure to gaseous pollutants were not showing statistically significant alteration.  However, result of single combined exposure (DEP+ gaseous pollutants) has been presented to evaluate the combination effect of DEP+ gaseous pollutant. Findings of all these different exposures mediated inflammatory/oxidative stress response has been discussed in detail in discussion section (lines 489-504).

  1. In figure7, MMP9 expression of DEP exposure groups was statistically significant different with control groups in “24 hours” and “72 hours”, but not 48 hours. Could you explain the finding? Similarly, in figure 5, the up regulation of HMOX1 and SOD3 was observed in “48 hours” groups, but not “72 hours” groups. Please explain the findings.

We have used PBECs from 3-4 donors (N) with three technical replicates for each to develop the bro-ALI (bronchial) models. ALI models are well-differentiated tissue-like models, which contain multiple layers of cells including different cell types of unique distribution. Hence, in this study, every model (ie. bro-ALI) can be recognized as a unique in vivo-like in vitro model with its own distribution of different cell types. Therefore, our experimental set-up includes both variation due to different donors as well as variation due to different cell distributions within each well. Thus, each trans well can be considered as an independent exposure. Additionally, since we have used different donors (3-4) for the primary bronchial epithelial cells (PBEC), each donor has very different basal levels of many of the markers. This is common and we have consistently observed this phenomenon also in relation to other previously published work of ours. This might the plausible cause for the variation observed in MMP9 expression in 3 different time points following exposure. Additionally, the unaltered TIMP, which is an inhibitor of MMP9 further support the alteration of extra cellular matrix homeostasis following repeated exposure to DEP at 48 h (lines 502-503). Figure 4e exhibit significant induced expression of HMOX1 at 48h following DEP exposure, which is highly comparable with reduced expression on MMP9. These findings are supported by several researchers (e.g.: Andrade et al. 2015, PMID: 26268658) showing that HMOX1 plays a central role in the regulation of the anti-apoptotic protein MMP-9 expression. It has been also well-established induced HO-1 activity, inhibits MMP-1 expression by suppressing c-Jun/AP-1 activation. These findings reveal a mechanistic link between oxidative stress and tissue remodeling which is comparable with the findings of the present study. We have included these in the discussion section (502-509).

  1. The inflammatory markers chosen in the DEP group were IL-8, IL-6 and TNF-α, but the markers used in the “DEP +gaseous pollutant” groups were CXCL8 and CXCL6. Why didn't they use the same markers and compare them together?

Thanks for identifying the error, these were typographical error. We have measured IL-8, IL-6 and TNF-alpha following all exposure studies. We have corrected the text and in figures accordingly.

Minor concern

  1. The figures should label as “A”, “B” and so on if there were more than one figure in “figure 4”, “figure 5” and so on.

we have included “a”, “b” and so on in each figure.

  1. CXCL8, CXCL6 were up regulated in “DEP + gas” group. However, I did not read the relative discussion. Please add some discussion.

IL6 expression was up-regulated following single combined exposure (DEP+Gas), which we have mentioned in the result section (lines: 386-387) though it was not statically significant. While IL8 expression remains unchanged following single combined exposure (Figure 7a). The markers which were showing statistically significant alteration following exposure, we have discussed in detail as well validated further with relevant or similar observation reported in other published studies.  

  1. Please explain why you select DEP as “particle” exposure, not PM collected from road or industrial area?

The primary reason for using DEP instead off PM collected from road or industrial area are as below:

1: The DEP used in this study was previously used in another study and results were already published by Ji et al, 2018. In this article, three different doses of DEP had been tested and its uptake followed analysis of cytotoxicity as well other molecular toxicity endpoints were also reported. Additionally,  Ji et al 2018 study distribution of DEP over the cells surface, agglomeration formation using PreciseInhale aerosol generator exposure system had been reported. All these information’s was useful and used as background data to set the correct dose of particles to perform the repetitive exposure in the present study.

2: Furthermore, this well characterized DEP with all background data from Ji et al, 2018 were easily available to perform all the required exposure for the completion of the present study. Hence, we have decided to perform this repetitive particle exposure study with our well characterized easily available DEP particle, rather than using any other particles (PM collected form road or industrial are), which always need significant time for the optimization of dose and time of exposure.

Round 2

Reviewer 2 Report

No further comment.